# On-Board Parameter Optimization for Space-Based Infrared Air Vehicle Detection Based on ADS-B Data

Yejin Li [1,2,3], Peng Rao [2,3], Zhengda Li [2,3] and Jianliang Ai [1,*]

1. Department of Aeronautics and Astronautics, Fudan University, Shanghai 200433, China; liyejin@mail.sitp.ac.cn
2. Shanghai Institute of Technical Physics, Chinese Academy of Sciences, Shanghai 200083, China; peng_rao@mail.sitp.ac.cn (P.R.); lizhengda@mail.sitp.ac.cn (Z.L.)
3. Key Laboratory of Intelligent Infrared Perception, Chinese Academy of Sciences, Shanghai 200083, China
* Correspondence: aijl@fudan.edu.cn

**Abstract:** Frequent aviation safety accidents of civil aircraft misses and crashes lead to an urgent need for flight safety assurance. Due to long-time flights over different backgrounds, accompanied by the changes in flight altitude and speed, it is difficult for a conventional space-based infrared detection system to use a set of fixed parameters to meet the stable detection requirement. To enhance the awareness of civil aircraft surveillance, a real-time parameter optimization method based on Automatic Dependent Surveillance-Broadcast (ADS-B) data is proposed. According to the background spectral characteristics and the real-time flight data, the most reasonable spectral band is analyzed, using the joint signal-to-noise/clutter ratio ($JSNCR$) as the evaluation criteria. Then, an automatic parameter adjustment is used to maximize the integration time and switch the integration capacitor gear. Numerical simulation results show that the $JSNCR$ increased by 1.16 to 1.31 times, and the corresponding noise equivalent target radiant intensity ($NET$) reduced from 2.4 W/Sr to 1.2 W/Sr compared with a conventional fixed-parameter detection system. This study lays a solid theoretical foundation for the spectral band analysis of space-based AVD system design. Meanwhile, the proposed method can be used as a standard procedure to improve on-board performance.

**Keywords:** civil aircraft surveillance; air vehicle detection; spectral band optimization; on-board parameter optimization

## 1. Introduction

The importance of air traffic for increasing the mobility of the population of the served regions is undeniable. Regular air transport is important for industry and logistics, but also for GDP growth [1]. The global airline industry conducted over 36.8 million flights worldwide in 2017, carrying over four billion passengers on scheduled flights. In recent years, frequent aviation safety incidents such as the MH370 crash have caused tremendous economic and life losses and revealed the necessity and strong demand for effective real-time aircraft monitoring on flight routes. The next generation of safety challenges now require the development and understanding of new forms of data to improve safety in other segments of commercial aviation, and moving from a reactive, incident-based approach toward a more proactive, predictive and systems-based approach [2].

Automatic Dependent Surveillance-Broadcast (ADS-B) is a multiparameter surveillance system designed to improve key segments of air traffic: enabling real-time surveillance, raising safety and efficiency levels and improving flight information and weather services [3]. ADS-B data generally contain information such as latitude and longitude, barometric altitude, GPS altitude, speed, heading and aircraft code. Satellites equipped with communication components can receive ADS-Bs in real-time and obtain target flight altitude and speed.

In addition to ground-based radar and airborne unmanned vehicle surveillance systems, a space-based surveillance system has great potential for real-time monitoring and wide-range coverage [4]. Space-based air vehicle detection (AVD) systems are based on passive radiation detection and do not require an active response from the target. Hence, they are an effective means of dynamic, global, real-time route monitoring. Under the conditions of space-based observation, it will be affected by earth background clutter, sunlight conditions, atmospheric absorption attenuation and flight status comprehensively. Due to the day and night difference in solar radiation, the radiant intensity of civil aircraft can be reduced by nearly 70% at nighttime, and the difference in background radiance can also decrease by 10 times in the short-wave infrared band. For the above reasons, it is difficult for the conventional space-based infrared detection system to use a set of fixed parameters to meet the stable detection requirement based on different scenarios. Additionally, the time efficiency of the traditional adjustment of detection parameters by sending commands from the ground is too low for time-sensitive target detection such as aircraft. Therefore, the research on reasonable spectral band selection and real-time adjustment of detection system parameters is of great significance.

Many researchers have made progress in the following aspects. In the aviation safety aspect, Xavier et al. [5] used the open source of ADS-B-based aircraft trajectories, and extracted information to present a framework to detect, identify and characterize anomalies in the past. Karl et al. [6] proposed a space-based ADS-B system and compared the strengths and weaknesses of potential space-based ATS systems. It has an advantage over traditional ground-based systems because it avoids the costly installation and maintenance of ground-based infrastructure while increasing the system's complexity at the same time. Gianluca et al. [7] integrated a traffic detect and avoid system into remotely piloted unmanned vehicles. The achieved results highlight the appropriate situational awareness provided by the proposed function and its effective support for remote pilots in making adequate decisions in conflict solving.

In modeling and experimental measurements on the infrared radiation of military and civil air vehicles, Mahulikar et al. and Eric et al. [8–10] presented a numerical method based on the finite element method to predict an aircraft's infrared signature. The method accounted for the aircraft's geometry, material properties, engine exhaust, flight conditions and detector position. Zhu et al. [11,12] established an all-attitude motion model for mid-wave and long-wave infrared space-based detection scenarios; the optimal detection spectrum can be selected through the signal-to-interference ratio. Bai et al., Lee et al. and Kou et al. [13–16] established an infrared radiation calculation model of the aircraft exhaust system and body according to the infrared radiation characteristics of aircraft. Considering the atmospheric attenuation along the radiation transmission path and the background noise received by the detector, the infrared characteristics of aircraft under different working conditions were analyzed. Ni et al. [17] studied the detection spectrum optimization problem of stealth aircraft targets on space platforms. By establishing a radiation model of stealth aircraft targets and backgrounds, the impact of different infrared bands on the detection performance of stealth aircraft targets was analyzed and the optimal detection band range was given. Deepti et al. [18] proposed a method based on artificial neural networks to identify the most useful spectral range for target discrimination. Shang et al. [19] proposed a target-constrained interference-minimized band selection method to improve the performance of target detection in hyperspectral images. The method uses the spectral information of targets and backgrounds, as well as the characteristics of target detectors, to select the optimal subset of spectral bands that maximizes the signal-to-noise ratio of target detection while minimizing interference. Yuan et al. [20,21] combined cloud top temperature, cloud pattern, cloud top height and other meteorological satellite sensor data to generate three-dimensional atmospheric transmittance and atmospheric path thermal radiation texture under different cloud heights and cloud phases. The results showed that the detection ability in the narrow bands of 2.65~2.90 μm and 4.25~4.50 μm was better than that in the broadband band of 3~5 μm. In the aspect of aerial target

detection and tracking, infrared image sequences from different platforms are important input sources for downstream target detection and tracking algorithms, and image quality directly determines the final detection and tracking performance. Several studies have carried out the evaluation of an enhanced YOLO target tracking algorithm based on infrared images obtained by a UAV platform [22–27].

Most of the previous studies focus on the infrared characteristic band selection to increase the contrast of the target and background, thereby improving the detectability of the system. To the best of our knowledge, existing studies have established spectral band selection methods based on infrared detection models and background clutter, but the detection scene changes of the daytime and nighttime differences are less considered, and the influence of different cloud backgrounds is not completely justified. The flight mode changes of air targets, such as reduced thrust takeoff or changes in cruising altitude and speed, have not been fully considered.

This paper proposes an on-board parameter optimization method based on prior ADS-B information to solve the above problems. The infrared signatures of multi-phase air targets and marine cloud backgrounds are analyzed. A space-based AVD model is established and the system performance is evaluated. The numerical simulation results show that the proposed method can effectively improve the detection robustness in different flight phases and backgrounds compared with the traditional fixed-parameter system. Our method can be generally applied to most space-based infrared systems for in-orbit performance improvement; meanwhile, it provides theoretical support for a new generation detection system of in-orbit adaptive parameter adjustment.

## 2. Methodology

A flow chart of an on-board parameter optimization method based on ADS-B is shown in Figure 1. Firstly, the typical infrared signature of civil aircraft was simulated combined with different marine cloud backgrounds. Next, a space-based infrared detection model was established and the system detection band was optimized using signal-to-noise ratio ($SNR$), signal-to-clutter ratio ($SCR$) and the joint signal-to-noise/clutter ratio ($JSNCR$) as the evaluation standard. Then, an automatic parameter adjustment algorithm was used to maximize the integration time based on real-time ADS-B flight speed. Finally, a switch to a more suitable integration capacitor gear was executed so that the noise was further suppressed. The output parameter set contained the optimized spectral band, the integration time and the integration capacitor gear.

### 2.1. Spectral Characteristics of Civil Aircraft with Multiple Flight Phases

The research on the infrared spectral characteristics of air targets is the basis of space-based infrared detection. Air vehicles generate infrared radiation during takeoff, cruise and landing, which can be detected using infrared detectors. The infrared radiation of the aircraft in the atmosphere mainly includes the infrared radiation of the high-temperature exhaust gas emitted by the engine, the skin radiation caused by aerodynamic heating and the scattering of the aircraft skin to the ambient infrared radiation. As the key component of an aircraft, the engine is the equipment to conduct the flight process, and its high-temperature exhaust plume generates a greater proportion of infrared radiation. The exhaust plume emitted from the engine is a selective radiator, and its infrared radiation is significantly higher in the carbon dioxide absorption spectrum, near 4.3 μm. In addition, when aircraft are flying at a high speed, the temperature rise of the fuselage due to aerodynamics also produces infrared radiation.

The main parameters that limit the maximum thrust of the engine are the speed of the turbofan, the pressure at the inlet of the combustion chamber and the temperature at the inlet of the turbine. Since the turbine inlet temperature is not easy to measure, the Exhaust Gas Temperature (EGT) was introduced. A high EGT is the main cause of engine repair. Therefore, most civil airliners adopt a thrust-reduced take-off mode [28]. The thrust-reduced take-off methods can be further divided in two: Derated Take-off and

Flexible Take-off Thrust. The specific implementation of the two methods is not the main focus of this paper, but the EGT result of thrust reduction can be used as the modeling input for calculating the infrared spectral characteristics of civil aircraft during take-off. That is, according to the regulations of the Federal Aviation Administration (FAA), when using Derated Take-off, the maximum thrust reduction cannot exceed 40% of full thrust; when using Flexible Take-off Thrust, the maximum thrust reduction cannot exceed 25% of the maximum take-off thrust. According to Ref. [29], a typical EGT can reach 783 °C (1056 K) during take-off at an external temperature of 15 °C, while reducing to 727 °C (1000 K) under 25% reduced thrust take-off mode.

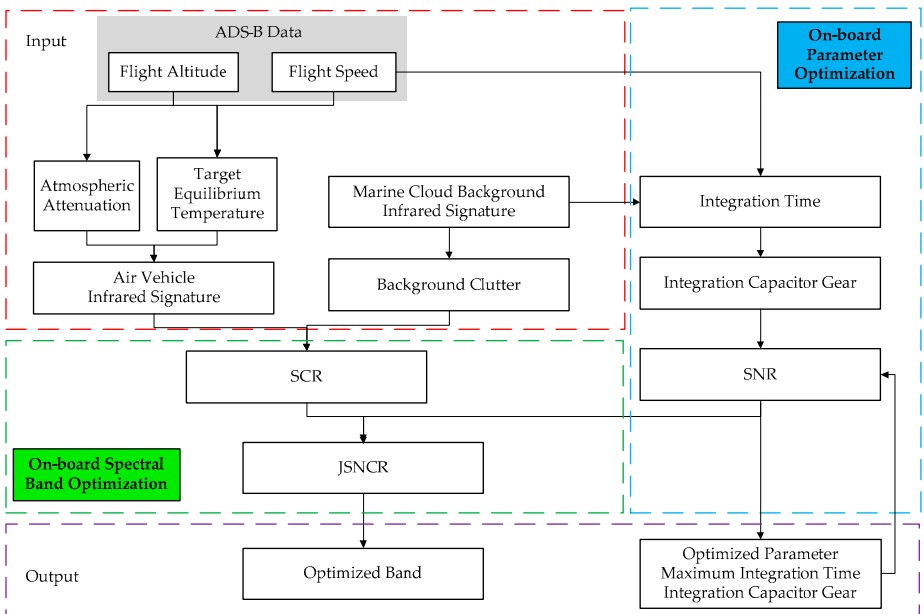

**Figure 1.** Flow chart of on-board parameter optimization method based on ADS-B. The inputs (in the red dashed box) are the target and background infrared signatures and ADS-B data (including flight altitude and speed); the on-board optimization includes two parts (in the green and blue dashed boxes), which are on-board spectral band optimization and on-board parameter optimization, using signal-to-noise ratio ($SNR$), signal-to-clutter ratio ($SCR$) and the joint signal-to-noise/clutter ratio ($JSNCR$) as the evaluation standard; the outputs (in the purple dashed box) are the optimized parameter set.

The infrared radiant intensity of an aircraft engine plume can be expressed by

$$I_{plume} = \varepsilon_{plume} \frac{M\left(T_{plume}, \lambda\right)}{\pi} A_{plume},$$ (1)

where $T_{plume}$ is the temperature of the exhaust plume, which can use EGT to characterize different flight phases; $\varepsilon_{plume}$ is the emissivity of the hot plume gas; $M\left(T_{plume}, \lambda\right)$ is the spectral radiant emittance of the plume; $\lambda$ is the spectral wavelength; and $A_{plume}$ is the projected area considering the detection angle and occlusion.

Civil aircraft generally fly at a cruise speed of 0.8 Ma, and the cruise altitude is usually greater than 6000 m. The best fuel economy altitude is generally around 9000 m. Aircraft skin will generate aerodynamic heating when flying in the air, and its skin stagnation point temperature $T_{skin}$ can be expressed as

$$T_{skin} = T_0\left[1 + k\left(\frac{\gamma - 1}{2}\right)M^2\right],$$ (2)

where $T_0$ is the ambient temperature; $k$ is the recovery factor; $\gamma$ is the gas constant; and $M$ is the flight Mach number.

In addition, in order to reduce the fuselage temperature during daytime flight, civil aircraft generally use white paint with a relatively high reflectivity [30]. This makes the solar scattering strong during daytime, which cannot be ignored. Therefore, the total infrared radiant intensity of the airframe can be expressed as

$$I_{skin} = \varepsilon_{skin} \frac{M(T_{skin}, \lambda)}{\pi} A_{skin} + r_{skin} \frac{S_{sol} \cdot \left( \vec{N} \cdot \vec{S} \right)}{\pi} A_{ref}, \tag{3}$$

where $T_{skin}$ is the temperature of the airframe, $\varepsilon_{skin}$ and $r_{skin}$ are the emissivity and reflectivity of the airframe, respectively, $M(T_{skin}, \lambda)$ is the airframe spectral radiant emittance, $A_{skin}$ is the projected area considering the detection angle and occlusion, $S_{sol} \cdot \left( \vec{N} \cdot \vec{S} \right)$ is the effective solar irradiance on the airframe and $A_{ref}$ is the effective area of the aircraft skin reflecting solar radiation. In certain spectral bands and applications, the impact of sky and earth radiation must be considered; we ignore these terms in this paper.

### 2.2. Electro-Optical Performance Model of a Space-Based Infrared System

The infrared detection system mainly depends on the temperature difference between target and background from the image sequence to realize detection, recognition and tracking. The core technical difficulty of AVD is to extract moving targets in a complex background. In recent years, large-scale staring infrared focal planes have become the mainstream of moving target detection systems due to their temporal and spatial advantages. Such systems can form wide-area coverage with large-scale detectors and achieve continuous observation through high frame rates, which meets the requirements of moving air targets better.

The sensitivity of the AVD system can be described by a series of noise equivalent sensitivity coefficients. The noise equivalent sensitivity coefficient is the smallest noise step that one system can distinguish, and it is represented at various positions along the imaging chain [31]. System sensitivity generally refers to the ability of the system to detect targets, and is usually defined as the change in radiation power when the output $SNR$ of the system is 1. As shown in Figure 2, at the target and background level we have noise equivalent target radiant intensity ($NET$), clutter equivalent radiant intensity ($CET$) and noise equivalent temperature difference (NEDT); at the aperture through atmosphere transmission, there is system-level noise equivalent irradiance ($\text{NEI}_{sys}$); at the surface of the focal plane, there is a detector-level noise equivalent irradiance ($\text{NEI}_{det}$); after the electro-optical conversion, there is noise equivalent charge ($NEC$); at the signal processing level, there is noise equivalent voltage (NEV); and finally, at the image level, we have noise-equivalent gray-scale (NEG). In this paper, two levels of noise equivalent sensitivity coefficients are highlighted, which are $NET$ and $CET$, and $NEC$.

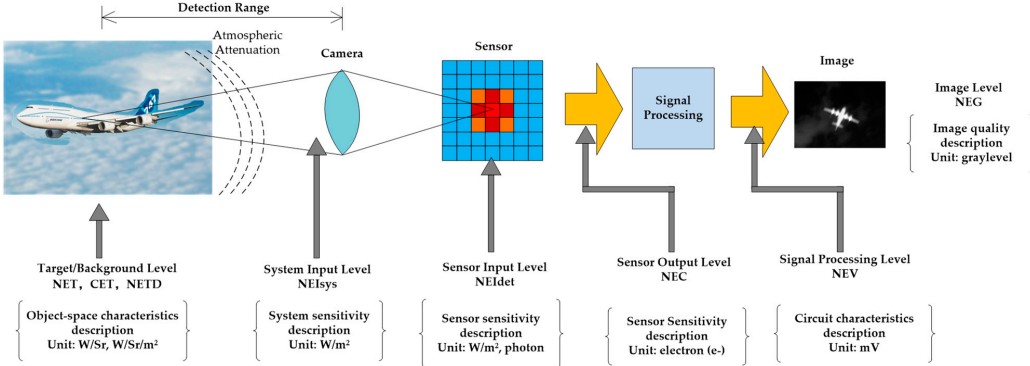

**Figure 2.** Noise equivalent sensitivity coefficient along the imaging chain.

### 2.2.1. Target Signal Model

The orbit height of satellites is generally hundreds of kilometers in low orbit and up to 36,000 km in high orbit, while the flight altitude of aircraft is generally around 10 km. Since the size of an air vehicle is generally smaller than the ground sample distance (*GSD*) of the AVD system, it can be considered as a point target [17]. Due to the diffraction effect of the electro-optical system, the point target was represented as a Gaussian distribution, in which the central pixel obtains most of the energy of the target, and generally forms a $3 \times 3$ point target by mixing with the background. The AVD system collected the infrared radiation, converted it into electrons and stored them in the integrating capacitor. The number of electrons received by the target pixel can be expressed as

$$N_s = QE \cdot \frac{\lambda_0}{hc} \cdot \frac{J\pi D^2 \tau_0 \tau_a EE}{4l^2} \cdot T_{int}, \tag{4}$$

where $QE$ is the quantum efficiency of the detector, $h$ is Planck's constant ($6.6 \times 10^{-34}$ J·s$^{-1}$), $\lambda_0$ is the center wavelength, $c$ is the speed of light, $\tau_a$ is the atmospheric transmittance at the height of the target, $\tau_0$ is the transmittance of the optical system, $J$ is the target radiant intensity, $l$ is the detection range and $D$ is the effective optical aperture diameter. $EE$ is the encircled energy of the optical system and $T_{int}$ is the integration time.

### 2.2.2. Noise Model

The system performance was limited by noise, which was characterized by temporal noise and spatial noise. Temporal noise represented the sensitivity of the payload itself, which can be expressed as the root mean square (RMS) of the electron collected by the integrating capacitance, also known as *NEC*. The equation of *NEC* is given by

$$NEC = \sqrt{n_{ph}^2 + n_{read}^2}, \tag{5}$$

where $n_{read}$ is the RMS readout noise and $n_{ph}$ represents the RMS noise of the sum of all photon noise, which can be divided according to different radiation sources into

$$n_{ph} = \sqrt{N_s + N_{bg} + N_{instr} + N_{dark}}, \tag{6}$$

where each term has units of electrons, $N_s$ is the photon signal, $N_{bg}$ is due to atmospheric path radiance and earth background radiance, $N_{instr}$ is the thermal emission from the optical elements and $N_{dark}$ is due to dark current generation.

When it is a point target detection scenario, the system sensitivity *NET* can be expressed as:

$$NET = \frac{NEC \cdot \frac{hc}{\lambda_0} l^2}{A_0 \tau_0 \tau_a QE \cdot T_{int}}, \tag{7}$$

where $A_0$ is the effective area of the optical system.

Since the earth clutter characteristics are related to spectral band, *GSD*, atmospheric transmission and background type, they vary dramatically with the observation scenario. Assuming that the total radiance of the earth clutter is a normal distribution, *CET* can be expressed as the standard deviation of the earth background fluctuation [32,33], which is given by

$$CET = \sigma_{clutter} \cdot GSD^2 \cdot Rad_{bg}, \tag{8}$$

where *GSD* is ground sample distance, $Rad_{bg}$ is background radiance and $\sigma_{clutter}$ is the background fluctuation as a percentage. *NET* and *CET* unify the performance unit to radiant intensity (W/Sr), which is consistent with the object space. Under the condition that the detection range remains unchanged, the normalization of the detection efficiency analysis can be realized. Since the spatial and temporal components are independently

decoupled, it simplifies the complexity of system performance evaluation. *JSNCR*, which combines temporal noise and spatial noise terms in radiant intensity, is expressed by

$$JSNCR = \frac{K \cdot EE \cdot J}{\sqrt{NET^2 + CET^2}},$$ (9)

where *K* is the target cross-pixel factor caused by the relative position change of target and satellite during observation period.

### 2.3. On-Board Parameter Optimization

#### 2.3.1. On-Board Spectral Band Optimization

The proposed method based on public ADS-B information can adjust the detection parameters more accurately and maximize the detection efficiency. Compared with the conventional radiation model calculation based on fixed flight height and speed, it is more accurate. During the flight, the spectral band is traversed to calculate the optimal band under different backgrounds. According to the characteristics of the target and the background, *JSNCR*, which combines *SNR* and *SCR*, is carried out in this paper as the criteria to perform spectrum traversal optimization. It is expressed as

$$\begin{cases} SCR(\lambda, \Delta\lambda) = \frac{K \cdot EE \cdot J(\lambda, \Delta\lambda)}{CET(\lambda, \Delta\lambda)} = \frac{K \cdot N_s(\lambda, \Delta\lambda)}{\sqrt{[\sigma(\lambda, \Delta\lambda)N_{bg}]^2}} \\ SNR(\lambda, \Delta\lambda) = \frac{K \cdot N_s(\lambda_{low}, \Delta\lambda)}{\sqrt{N_s(\lambda, \Delta\lambda) + N_{bg}(\lambda, \Delta\lambda) + N_{instr}(\lambda, \Delta\lambda) + N_{dark} + N_{read}^2}} \\ JSNCR \cdot (\lambda, \Delta\lambda) = \frac{K \cdot N_s(\lambda, \Delta\lambda)}{\sqrt{N_s(\lambda, \Delta\lambda) + N_{bg}(\lambda, \Delta\lambda) + N_{instr}(\lambda, \Delta\lambda) + N_{dark} + N_{read}^2 + [\sigma(\lambda, \Delta\lambda)N_{bg}]^2}} \end{cases},$$ (10)

where λ is the lower band edge of the spectral band and Δλ is the bandwidth.

The threshold of different criteria may vary according to the actual detection scenario: *SCR* threshold needs to be considered in strong clutter and weak clutter. Generally, *SCR* > 10 can be considered as an ideal weak clutter. When *SCR* is less than 1, it is considered to be a strong clutter scene. In this case, the clutter dominated the final detection performance [34,35]. Since there was no background clutter greatly affecting detection at nighttime, the analysis focused on the optimization of a given background band that was easily affected by strong stray radiation during the daytime. In this paper, we set the clutter ratio σ as 5% during the daytime.

The *SNR* threshold was generally set to 6 for target detection and 3 for target tracking. The target recognition was better than 30 [36]. *JSNCR* reflects the influence of both *SNR* and *SCR* at the same time; the *JSNCR* optimization problem in the cruising phase is similar to that of the target tracking process. Considering the weak infrared signature of air targets at nighttime and the current status of multi-frame correlation for moving target detection, the final *JSNCR* threshold was set to greater than 3 [37].

The on-board parameter optimization flow during the detection can be divided into following four main steps.

First of all, build the detection scenario according to a typical civil air target flying above a marine background with clouds; initialize the main parameters such as the detection band, integration time and integral capacitance of the system; and calculate the current *SCR*, *SNR* and *JSNCR*.

Secondly, obtain the real-time flight state data from ADS-B, including flight altitude and speed, combined with the background spectral characteristics around the target area.

Then, set (λ, Δλ) as variables, where λ starts from 2.0 to 4.0 μm and Δλ is from 0.1 to 0.5 μm. Calculate the *JSNCR* under a specific background, and the (λ, Δλ) corresponding to the obtained maximum *JSNCR* value is the result after spectrum optimization.

Finally, adjust the most reasonable detection band for the current scenario.

### 2.3.2. On-Board Parameter Optimization

In the process of detection parameter optimization, under the condition that the detection range remains unchanged, the integration time was the major influencing factor, and it was positively correlated with the system performance; that is, $SNR$ was proportional to the square root of the integration time.

For the AVD system, there were two aspects that limited the maximum integration time: one was the time for the moving target to fly out of one pixel (the high-resolution imaging system may not exceed 1/4~1/2 pixel as the limit condition); the second was the time from the integration of the superimposed radiation energy of the target and the background to the saturation of the detector pixel. The minimum value of the two was the limit for the maximum integration time. Therefore, $T_{int\_max}$ is represented as

$$T_{int\_max} = min\left\{ \frac{N_{full}}{P_{max}}, \frac{GSD}{v_{tar}} \right\},\tag{11}$$

where $N_{full}$ is defined as the maximum amount of electrons that can be stored within an individual pixel. Generally, based on current detector readout circuit technology, one pixel can have two levels of integral capacitor, and can switch according to the different demands; $P_{max}$ is the electron flow density per unit time after the detection target is mixed with the background; and $v_{tar}$ is the real-time flight speed of the air target.

The on-board parameter optimization steps are described in Algorithm 1. Additionally, the detailed description is as follows:

---

**Algorithm** 1 On-board Parameter Optimization

---

Input: Parameter $N_{full}$, $P_{max}$, $v_{tar}$, default $T_{int\_0}$, default integral capacitor gear $C_0$
Output: Parameter maximum $T_{int\_max}$, matched integral capacitor gear $C_m$
1: Initialize $T_{int\_0}$, $C_0$.
2: Calculate $JSNCR \cdot (T_{int\_0}, C_0)$.
3: while ($JSNCR(T_{int\_max}, C_m) < JSNCR(T_{int\_0}, C_0)$) do
4:　Calculate $\frac{N_{full}}{P_{max}}$, $\frac{GSD}{v_{tar}}$.
5:　Set $T_{int\_max} = min\left\{ \frac{N_{full}}{P_{max}}, \frac{GSD}{v_{tar}} \right\}$.
6:　Match integral capacitor gear $C_m$.
7: end while
8: Update parameter $T_{int\_0}$ as $T_{int\_max}$, $C_0$ as $C_m$.

---

(a) Initialize the integration time $T_{int\_0}$ and the integration capacitance gear $C_0$ according to the default value to detect the air target;

(b) After the air target is detected, accumulate multi-frame images to obtain the background radiation characteristics of the current scene, and at the same time receive the ADS-B data of the civil target. Use the flight speed to fit the flight trajectory and process the motion speed estimation, integrate with the mixed target and background radiation characteristics of neighborhood pixel and calculate the maximum integration time $T_{int\_max}$;

(c) Match the integral capacitor gear $C_m$ according to $T_{int}$, and calculate $JSNCR(T_{int}, C_m)$;

(d) If $JSNCR(T_{int}, C_m) > JSNCR(T_{int\_0}, C_0)$, update the parameters;

(e) If the detection inputs such as flight speed and detection background change, steps (b), (c) and (d) are repeated.

## 3. Results and Discussion

### 3.1. Calculation Conditions

#### 3.1.1. Multi-Phase Aircraft Infrared Signature

A big commercial aircraft was used as the analysis target and its infrared signature during take-off and cruising phases was simulated. The skin material of the aircraft was white paint, the emissivity was 0.8 and the reflectivity was 0.2. When the thrust was reduced by 25% during the take-off phase, the initial skin temperature was 288 K and EGT

was 1000 K. During the cruising phase, the skin equilibrium temperature was 255 K, and the cruising altitude was 10 Km [37–39]. The results of the spectral radiant intensity in the two phases are shown in Figure 3.

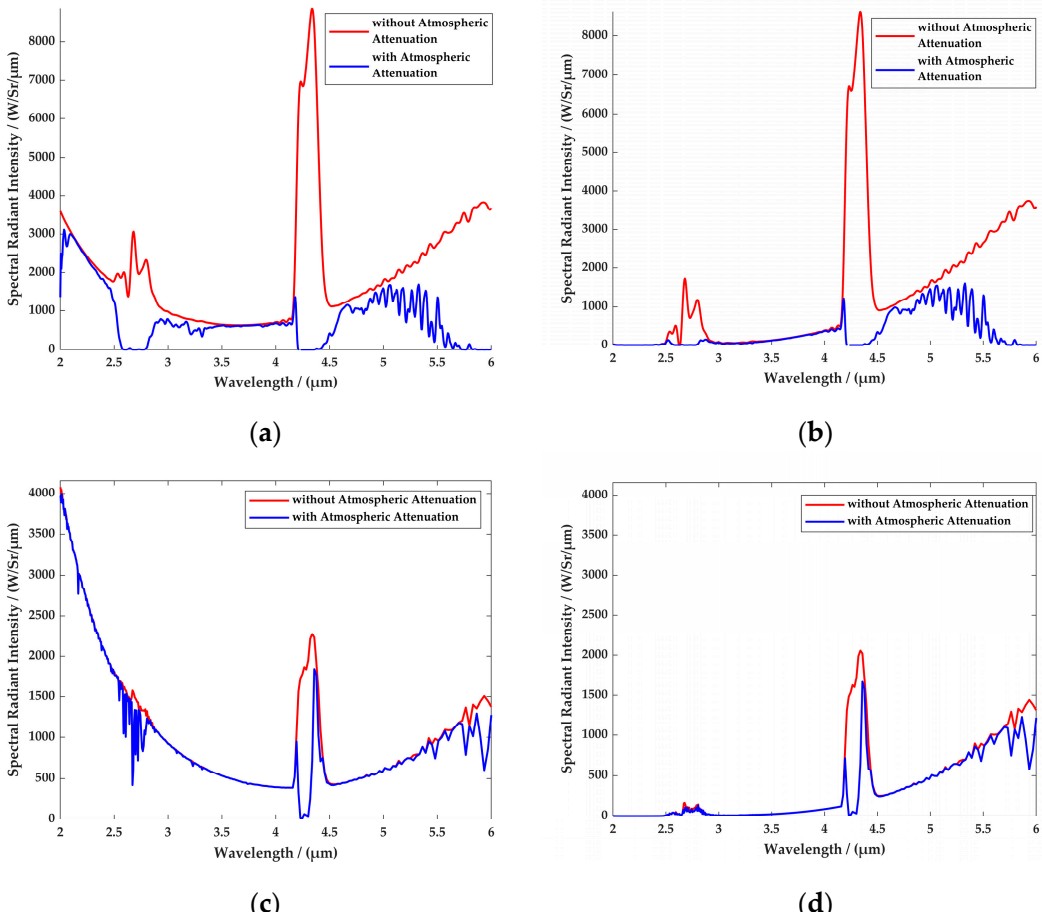

**Figure 3.** Big commercial aircraft spectral radiant intensity; (**a**) reduced-thrust takeoff (daytime), (**b**) reduced-thrust takeoff (nighttime), (**c**) cruise (daytime), (**d**) cruise (nighttime).

The flight route used for analysis in this paper was obtained from the measured Flight Operations Quality Assurance (FOQA) data. The FOQA flight data come from the real parameters of various instruments and equipment of the aircraft. When an aircraft is in flight, the FDIMU data collector on the aircraft collects thousands of high-quality original parameters on the aircraft bus per second according to the ARINC 429 protocol. The data are acquired in real time on-board the aircraft and downloaded by the airline once the aircraft reaches the destination gate [40,41]. The altitude and speed changes of the flight route are shown in Figure 4. The total flight time was about 1 h and 22 min, and its cruising time was about 2400 s. The detection optimization method in this paper focuses on the cruising phase, as shown in the red box in Figure 4.

The distribution of radiant intensity of a large commercial aircraft with different spectrum bands is shown in Figure 5. It can be seen that during the daytime, the r the target radiant intensity is mostly hundreds of W/Sr, and at nighttime without sunlight, the target radiant intensity is extremely reduced, even below 10 W/Sr.

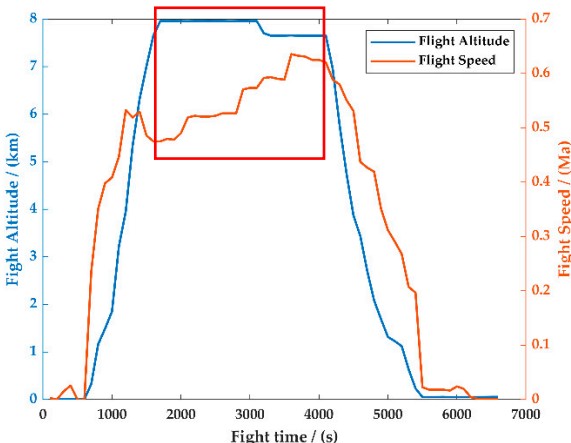

**Figure 4.** Flight speed and altitude of the airline, the cruising phase is highlighted with the red box.

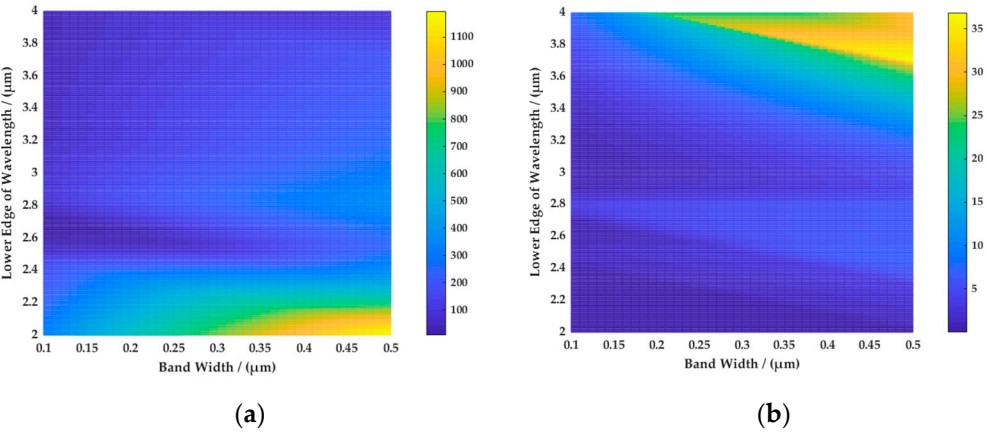

(**a**)  (**b**)

**Figure 5.** A commercial aircraft's radiant intensity (W/Sr). (**a**) Cruise (daytime), (**b**) cruise (nighttime).

### 3.1.2. Background Simulation

The spectral properties of different cloud marine backgrounds were simulated. In this study, MODTRAN 5 was used to simulate the atmospheric radiative transfer process and obtain the background radiance. The background types were sea, cumulus cloud, altostratus cloud and cirrus cloud in the equatorial region [42–44], and the calculation conditions were divided into daytime and nighttime. The simulation results are shown in Figure 6. It can be seen that the cloud background was strongly affected by sunlight during the daytime, which was the main factor affecting the detection.

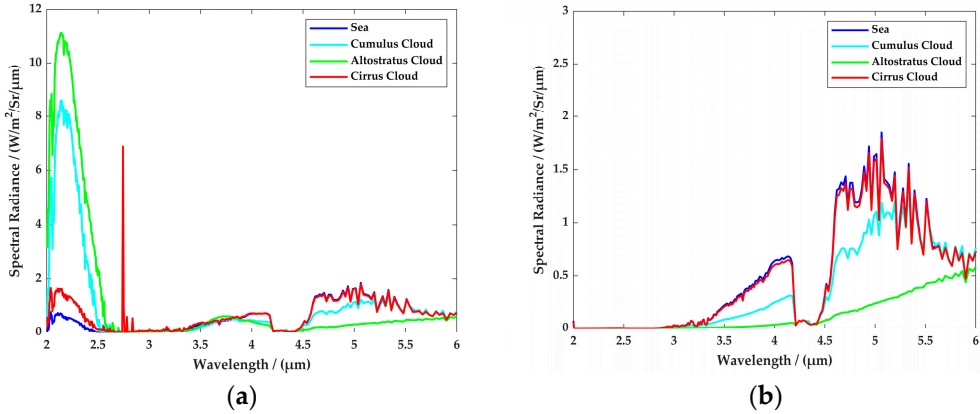

(**a**)  (**b**)

**Figure 6.** Background spectral radiance. (**a**) Daytime, (**b**) nighttime.

### 3.1.3. Space-Based AVD System

At present, the space-based infrared detection system is represented by OPIR and SBIRS of the United States; however, there is no space-borne infrared system specialized in AVD. According to the *GSD* need for a space-based AVD system [12] and a published article [21], the typical parameters of AVD system were estimated, which are listed in Table 1. The superiority of the proposed method is illustrated by comparing it with the traditional fixed-parameter AVD system.

**Table 1.** Space-based AVD system parameters.

| Parameter | Fixed Parameter | Proposed |
|---|---|---|
| Orbit altitude | 36,000 km | 36,000 km |
| Optical aperture | 2.3 m | 2.3 m |
| *GSD* | 70 m | 70 m |
| Detection band | 2.7–3.0 μm (fixed) | 2.0–4.5 μm, Dynamically adjustable |
| Integration time | 100 ms (daytime), 200 ms (nighttime) | 10 μs–1000 ms, Dynamically adjustable |
| Integration capacitance gear (full well) | $3 \times 10^6$ e$^-$, $6 \times 10^6$ e$^-$ (default $3 \times 10^6$ e$^-$) | $3 \times 10^6$ e$^-$, $6 \times 10^6$ e$^-$, Dynamically adjustable |

The spectral band dynamically adjusted from 2.0 to 4.5 μm, and the integration time and the capacitance gear were adjustable. The performance of the proposed method over different detection scenarios was compared with a fixed-parameter AVD system with a fixed-parameter set (Detection band: 2.7–3.0 μm, Integration time: 100 ms at daytime, 200 ms at nighttime, Default integration capacitance: $3 \times 10^6$ e$^-$).

The proposed method and other method were implemented under MATLAB R2019b with an Intel Core 2.80 GHz processor and 8 GB of physical memory. *JSNCR* results were analyzed after executing the parameter adjustment compared with the fixed-parameter system, and several conclusions and suggestions were achieved.

### 3.2. On-Board Spectral Band Optimization

The spectral band optimization results of *SCR* are shown in Figure 7. The extreme value was obtained near the 2.6–2.8 μm absorption band, and they were all higher than 100, except the daytime cirrus cloud background. The *SCR* value in (c) was the lowest; therefore, it was the most difficult to detect the target, and *SCR* obtained a peak value at 2.8 μm, which was above 60. According to the *SCR* threshold discussed in the previous section, it can be seen that most of the spectral bands selected based on the target and background contrast met *SCR* greater than 10, and the specific spectral band should be further determined by combining camera *SNR* to achieve *JSNCR*.

As shown in Figure 8, the spectral band optimization results of *SNR* show that the value varied greatly in different backgrounds and that there was no uniform trend, but the *SNR* was higher during the daytime than the nighttime. It was noted that SNR had a negative value under some nighttime conditions, which indicated that the energy of the air target was lower than that of the night background, showing a negative contrast. The optimal spectrum segment should be determined according to the *SNR* threshold value and combined with *SCR* using *JSNCR* as the final criterion.

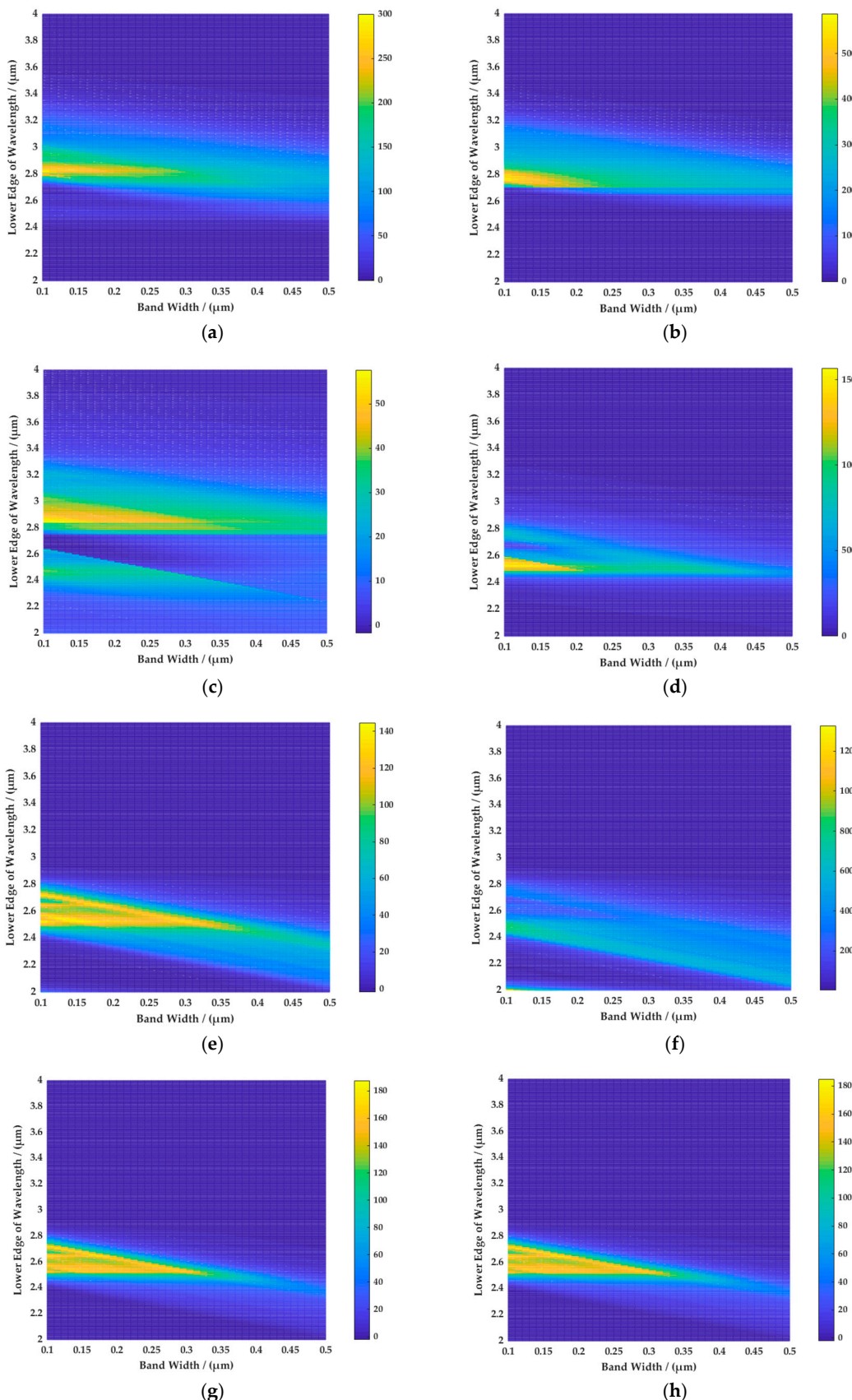

**Figure 7.** The spectral band optimization results of *SCR* with different backgrounds. (**a**) Daytime cumulus cloud; (**b**) daytime altostratus cloud; (**c**) daytime cirrus cloud; (**d**) daytime sea; (**e**) nighttime cumulus cloud; (**f**) nighttime altostratus cloud; (**g**) nighttime cirrus cloud; (**h**) nighttime sea.

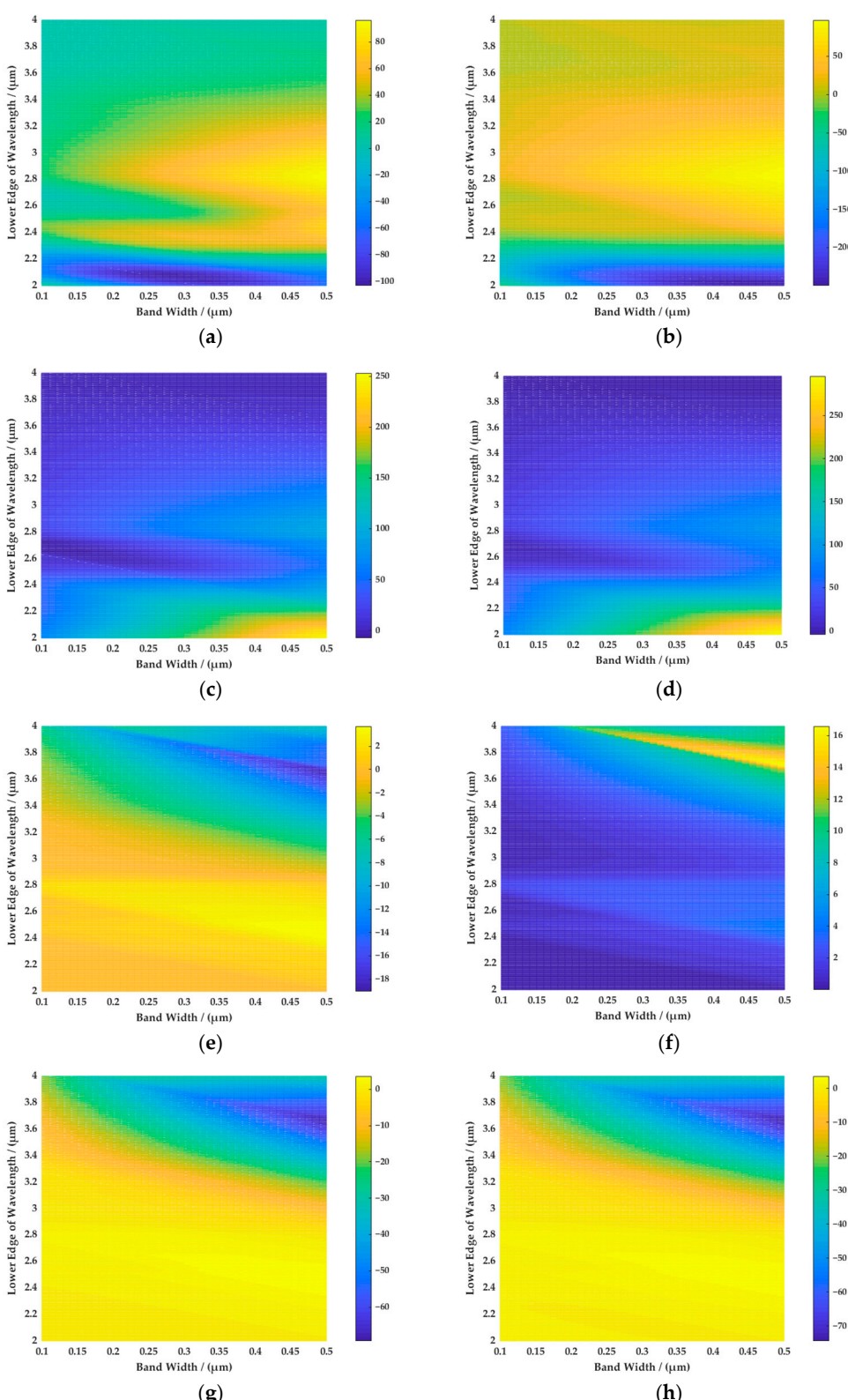

**Figure 8.** The spectral band optimization results of *SNR* with different backgrounds. (**a**) Daytime cumulus cloud; (**b**) daytime altostratus cloud; (**c**) daytime cirrus cloud; (**d**) daytime sea; (**e**) nighttime cumulus cloud; (**f**) nighttime altostratus cloud; (**g**) nighttime cirrus cloud; (**h**) nighttime sea.

Based on ADS-B data, the optimization spectrum band results of the four types of background through *JSNCR* are shown in Figure 9. It can be seen that in (a) and (b) the background optimization results of cumulus and altostratus were similar, and

the highest performance was obtained in the 2.822–3.244 µm range under the cumulus cloud background, which was about two times that of the fixed-parameter system; the altostratus cloud background obtained the highest *JSNCR* in the 2.787–3.287 µm range, about 1.7 times the detectability of the fixed-parameter system; the optimized spectral band of the cirrus cloud background in (c) was 2.842–3.112 µm, and the performance was significantly improved after spectrum optimization, which was more than six times higher than that of the fixed-parameter system; 2.45–2.95 µm under the background of the sea was the optimal detection band in (d), and the *JSNCR* was about two times that of the fixed-parameter system. Combining the common parts of (a), (b) and (c) intervals, taking 2.8–3.2 µm as the common detection band interval can meet all cloud marine background detection requirements. Under night detection conditions (e)–(f), 2.45–2.95 µm was the preferred band, but the *JSNCR* was lower and only greater than three. The overall optimization results are summarized in Table 2. It can be found that bringing in real-time flight data is closer to the actual detection scenario. *JSNCR* improved significantly during the daytime after the spectral band optimization; at the same time, it was also found that low *JSNCR* was common in the AVD during the nighttime.

For the cruising phase of the flight route in the previous section, its dynamic speed curve is shown in Figure 10. During the cruising period from 1700 s to 4100 s, the flight speed monotonously increased from 0.47 Ma to 0.62 Ma. Based on a *GSD* of 70 m and the limit of air target flying no more than one pixel, the maximum integration time was 433 ms to 331 ms, which matched the velocity curve, as shown on the right axis in the figure.

According to the result of spectral band optimization, the detection spectral band was preferably 2.8–3.2 µm during the daytime for the cloud backgrounds, whose radiance was from 0.088 to 0.03 W/Sr/m$^2$. As the current band selection was a narrow band, and the target radiation intensity was only in several hundreds of W/Sr, there was no need to consider the saturation due to the strong target, and the maximum integration time was mainly limited by the target flight speed The comparison between the fixed-parameter system spectrum of 2.7–3.0 µm and the optimized *JSNCR* performance improvement is shown in Figure 11a and Table 3. It can be seen that the overall performance was improved after adjusting the detection spectrum, especially for the cirrus cloud background. *JSNCR* increased about four times, greatly improving the detectability under strong background clutter during the daytime. Furthermore, based on the spectral band optimization, we performed a dynamic parameter adjustment of the maximum integration time and the matching of the integration capacitor gear, and the result is shown in Figure 11b and Table 3. It can be seen from the results that the parameter optimization can only improve the detection scenario dominated by non-background clutter, and had no obvious improvement on the performance of the cirrus cloud background, while for the cumulus and altostratus background with weaker background clutter, there was still a certain effect. The corresponding *JSNCR* increased by 1.16 to 1.31 times, and *NET* reduced from 2.4 W/Sr to 1.2 W/Sr.

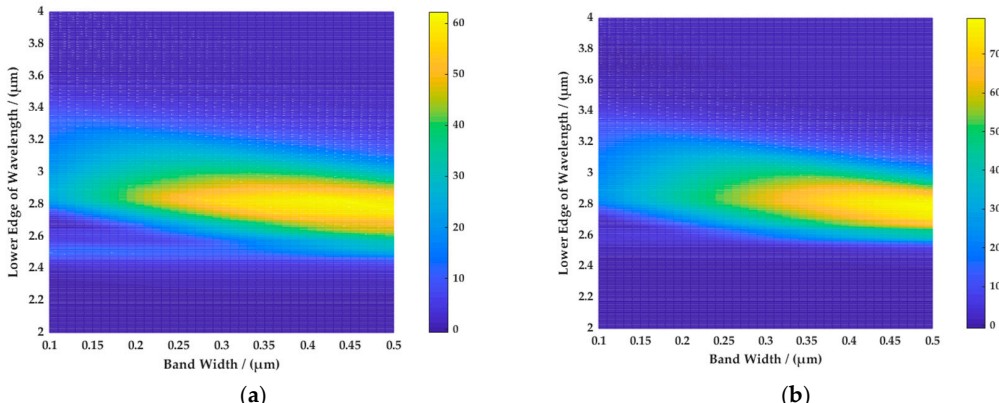

**Figure 9.** *Cont.*

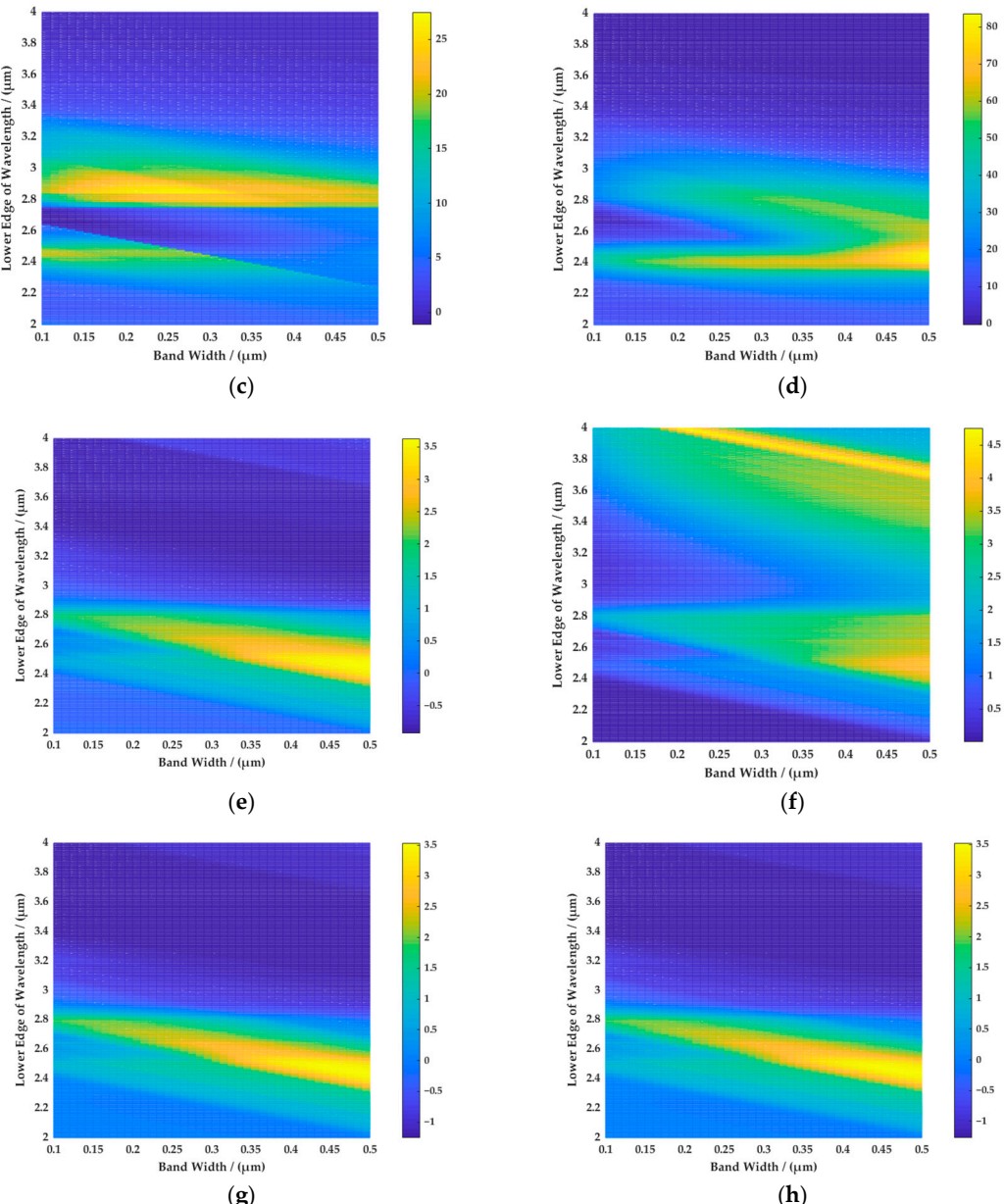

**Figure 9.** The spectral band optimization results of *JSNCR* with different backgrounds. (**a**) Daytime cumulus cloud; (**b**) daytime altostratus cloud; (**c**) daytime cirrus cloud; (**d**) daytime sea; (**e**) nighttime cumulus cloud; (**f**) nighttime altostratus cloud; (**g**) nighttime cirrus cloud; (**h**) nighttime sea.

**Table 2.** The optimization results of spectral band and *JSNCR*.

| *JSNCR* @Spectral Band (μm) | | |
|---|---|---|
| **Background** | **Fixed Parameter** | **Proposed** |
| daytime | | |
| Cumulus cloud | 36.47@2.7–3.0 | 61.03@2.822–3.244 |
| Altostratus cloud | 45.57@2.7–3.0 | 79.27@2.787–3.287 |
| Cirrus cloud | 4.18@2.7–3.0 | 27.49@2.842–3.112 |
| Sea | 43.75@2.7–3.0 | 83.56@2.45–2.95 |
| nighttime | | |
| Cumulus cloud | 2.38@2.7–3.0 | 3.63@2.45–2.95 |
| Altostratus cloud | 2.88@2.7–3.0 | 4.08@2.45–2.95 |
| Cirrus cloud | 1.95@2.7–3.0 | 3.47@2.45–2.95 |
| Sea | 2.04@2.7–3.0 | 3.47@2.45–2.95 |

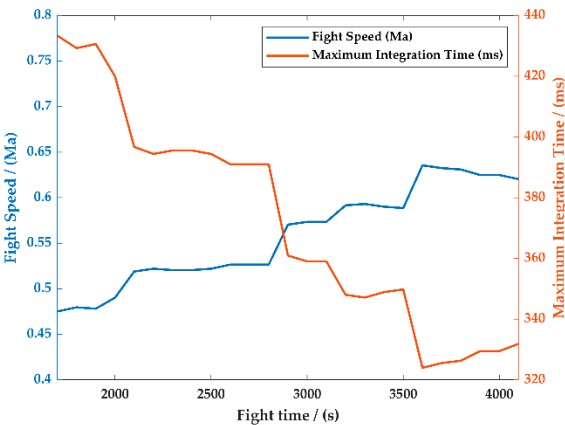

**Figure 10.** The maximum integration time during the cruising phase.

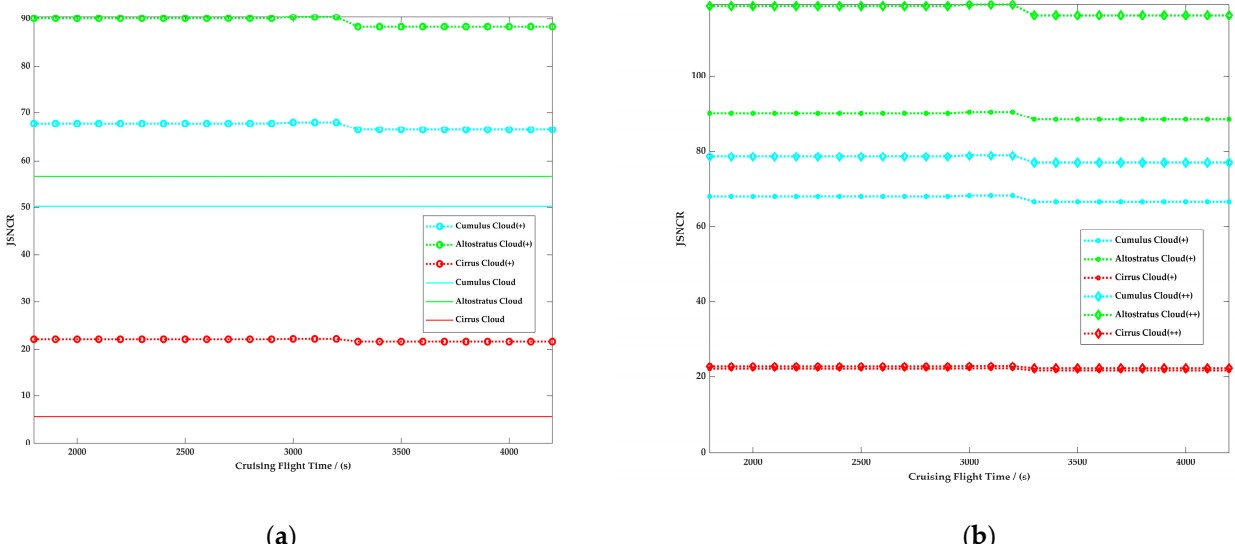

(**a**)                                                                  (**b**)

**Figure 11.** Performance improvement for cumulus cloud, altostratus cloud and cirrus cloud backgrounds (daytime). (**a**) The straight line represents *JSNCR* of the fixed-parameter system, and the straight line with circles represents *JSNCR* after the proposed on-board spectral band optimization algorithm; (**b**) the straight line with dots represents *JSNCR* after the proposed on-board spectral band optimization algorithm, the straight line with diamonds represents *JSNCR* after the proposed on-board spectral band optimization and on-board parameter optimization algorithm.

**Table 3.** The optimization result of *JSNCR* performance.

| | *JSNCR* | | |
|---|---|---|---|
| **Background** | **Fixed Parameter** | **Proposed On-Board Spectral Band Optimization** | **Proposed On-Board Spectral Band Optimization and On-Board Parameter Optimization** |
| Cirrus cloud | 5.66 | 22.02 | 22.68 |
| Cumulus cloud | 56.67 | 66.49–67.83 | 76.9–78.47 |
| Altostratus cloud | 50.21 | 88.37–90.42 | 115.88–118.22 |

In summary, the performance of the proposed method had a significant improvement compared with that of the traditional fixed-parameter system; especially in the cirrus cloud background, which had the greatest impact on AVD, the improvement was up to four times. Based on the improvement and maximization of *JSNCR* in the cruising phase, the on-board spectral band and parameter optimization process was carried out so that the

*JSNCR* was kept at a high level, and realized high-quality feature extraction for further target recognition. At the same time, compared with ground-based and airborne platforms, the advantage of the space-based AVD system is realizing multi-target real-time awareness and wider coverage. The high performance of target detection and tracking algorithms is very dependent on the raw infrared image signal-to-noise ratio [36]. The method proposed in this paper can be used for target detection and support for performance improvements in target tracking algorithms such as YOLOv5 [27].

### 4. Conclusions

In this paper, an on-board parameter optimization method based on ADS-B data is proposed. Parameters such as detection band, integration time and integration capacitance are adjusted in real-time to improve the target detection and recognition performance. Numerical simulation results show that the system performance after on-board parameter optimization using ADS-B data had a significant improvement compared with a conventional fixed-parameter detection system. The final optimized spectral band of marine cloud background was 2.8–3.2 μm during daytime, and 2.45–2.95 μm in the nighttime. The proposed method increased *JSNCR* by 1.16 to 1.31 times and reduced *NET* from 2.4 W/Sr to 1.2 W/Sr, which better meets the robust detection need of an air vehicle in complex cloud clutter. This study lays a solid theoretical foundation for the spectral band analysis of space-based AVD system design. Meanwhile, the parameter optimization can be used as a standard procedure to improve on-board performance.

In future work, the use cases of the algorithm for the downstream modules in terms of object detection and tracking using simulation infrared image sequences with the AVD scenario should be further carried out.

**Author Contributions:** Conceptualization, Y.L. and J.A.; Investigation, Y.L. and Z.L.; Methodology, Y.L. and P.R.; Simulation, Y.L. and Z.L.; Data curation, Z.L.; Funding acquisition, P.R.; Project administration, P.R. and J.A.; Supervision, J.A.; Writing—Original draft preparation, Y.L.; Writing—Review and editing, P.R. and J.A. All authors have read and agreed to the published version of the manuscript.

**Funding:** This research was funded by the National Natural Science Foundation of China, grant number 62175251. This research work was supported by the open project of the Key Laboratory of Intelligent Infrared Perception, project ID: CAS-11RP-02.

**Data Availability Statement:** Not applicable.

**Conflicts of Interest:** The authors declare no conflict of interest.

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
