# Peer review of "On-Board Parameter Optimization for Space-Based Infrared Air Vehicle Detection Based on ADS-B Data"

_applsci, doi:10.3390/app13126931_

Round 1

Reviewer 1 Report

The overall impression of the work is positive. However, there are a few comments:

1. The number of references to researches for 2018 – 2022 is less than 30% of the total number of sources.

2. The review provides an analysis of a number of approaches to solving the problem under consideration. However, their advantages and disadvantages are not indicated from the point of view of practical application and why further research on this topic is relevant.

3. It would be useful to present the results of a comparative analysis of the characteristics of the method developed by the authors with the characteristics of other methods given in the review.

4. The article presents the quantitative values of the method parameters. However, there is no clear justification for their choice.

5. The article lacks the characteristics of software and computing devices that were used for numerical modeling.

I believe that after taking into account these comments, the article can be recommended for publication.

Author Response

Dear Reviewer:

Thank you for your comments concerning our manuscript titled “On-board Parameter Optimization for Space-based Infrared Air Vehicle Detection based on ADS-B data”. We have revised the manuscript according to your comments.

We are very grateful for the thorough comments from you, as they greatly helped us to improve this letter. We have carefully considered the suggestions of the reviewers and hereby revised our text and figures accordingly. We hope that this version positively addresses your comments to meet your expectations.

The responses to your comments have been uploaded to the attachment.

Sincerely,

Yejin Li, Peng Rao, Zhengda Li, Jianliang Ai

 28.5.2023

Reviewer 2 Report

In this paper, to address the critical issue of aviation safety accidents or crashes and enhance the awareness of civil aircraft surveillance, a real time parameter optimization method based on automatic dependent surveillance broadcast data is presented. Based on the statistical results of background spectral characteristics and the real-time flight data, the most reasonable spectral band is analyzed, using the joint signal-to-noise/clutter ratio (JSNCR) as the evaluation criteria. Further, authors proposed an automatic parameter adjustment algorism to maximize the integration time based on the current flight speed. Last but not least, a switch to a more suitable integration capacitor gear is executed so that the noise is further suppressed. These propositions have been validated by the comprehensive numerical simulation results showing good performance. With that said, I would say this is a good piece of work that will contribute to the aviation industry regarding the safety issue and suggest an acceptance with some minor modifications. Below are my comments:

-For figure 1, please add more descriptions to the caption. The acronym at least should be explained. With more details, the figure can be more readable.

-Is that possible to have some benchmark work? Or use the results of this work to support other works such as object detection using image processing techniques. If not, that is ok, because the results in this paper have already verified optimization approaches.

-The paper has presented valuable research on finding the parameters for the spectral band. Based on these methodology and results, I suggest the authors adding some use cases of the algorithm for the downstream modules in terms of object detection and object tracking. Object tracking research in the literature: an automated driving systems data acquisition and analytics platform, and object detection research: yolov5-tassel: detecting tassels in rgb uav imagery with improved yolov5 based on transfer learning, should at least be discussed to elaborate the use cases and applications.

-I suggest having a pseudo code for the parameter optimization process for section 2.3.2.

Author Response

(The authors gave the same response as above.)

Reviewer 3 Report

Dear Authors,

The article deals with security systems in air transport. Frequent aviation safety accidents related to the disappearance and breakdown of civil aircraft create an urgent need to ensure the safety of aircraft flights. The authors present on-board optimization of parameters for detecting infrared air vehicles in space. These are very important issues that can improve safety in the civil aviation sector. 
The manuscript is generally well prepared, although it has some shortcomings:

First, the abstract is too long, it should contain up to 200 words.

Literature review, although the literature is well selected, it is too poor (only 22 items). The introduction should be expanded on the situation on the civil aviation market (especially small airports) and air transport safety. These issues are marginal in the manuscript and it should be supplemented. To help authors, I suggest adding, for example, these works:

https://doi.org/10.1051/matecconf/201823602007

Some figures (Fig. 6-8) and graphs (Fig. 3-5 and Fig. 9-11) are poorly visible, their quality should be improved.

In addition, there is a lack of discussion of the experiments with other scientific works in this area, which should be supplemented. Consider, for example, these works:

https://doi.org/10.3390/aerospace9100531

In addition, there are also minor editing errors, for example in the notation of units (2.45-2.95μm, there should be a space before the unit), especially in chapters 3 and 4.

The strength of the manuscript is a good description of the research methodology and the description of the results obtained (strengthen discussion of results).

Therefore, I believe that the manuscript has great potential and is worth publishing when completed.

Thank you

Author Response

(The authors gave the same response as above.)

Reviewer 4 Report

This manuscript describes an on-board parameter optimization method for space-based infrared air vehicle detection, on Automatic Dependent Surveil-lance-Broadcast (ADS-B) data with the joint signal-to-noise/clutter ratio (JSNCR) as the evaluation criteria. Some issues might be clarified:

1. Some additional clarifications might be briefly made in the Introduction section, for more general readers. For example, is the on-board processing expected to be done on the satellite? What are the reasons requiring on-board processing (e.g. raw sensor data cannot detect all relevant spectral bands simultaneously, etc.)?

2. In Section 2.1, it is first stated that "[aircraft] high-temperature exhaust plume is the main source of infrared radiation", but then later that "infrared radiation... mainly includes... high-temperature exhaust gas..., skin radiation... and scattering of aircraft skin", which is three components. The expected relative contribution of these three components to infrared radiation might be clarified.

3. T_s is defined in Equation 2, but does not appear to be used in Equation 3 for spectral characteristic calculation. This might be clarified.

4. In Section 2.2.1, it is stated that the size of the air vehicle is generally smaller than the ground sample distance of the AVD system, so it can be considered a point target. It might be clarified if the altitude of the flight is significant in relation to ground sampling distance.

5. In Section 2.3.1, typical thresholds for JSNCR (>3) etc. are mentioned. Further justification for such threshold values might be cited, as done for SNR [17].

6. In Section 3.3.1, it is stated that the take off and cruising infrared signature of commercial aircraft were simulation, but also later that the actual flight path of a civil airline was taken. More details might be provided about how such simulation (including for clouds later) was implemented, and where/how actual flight data was obtained from.

7. In Section 3.2, spectral band optimization results are shown in the figures. If possible/relevant, the aircraft signature might be identified too for better understanding.

Minor issues such as "algorism" in the Abstract

Author Response

(The authors gave the same response as above.)

Round 2

Reviewer 3 Report

Dear Authors,

thank you for making changes to the manuscript and considering my suggestions. The article is well revised and I recommend it for publication. You only need to adapt the literature to the journal template.

Thank you.

Author Response

Dear Reviewer,

Thank you for your new comments concerning our manuscript titled “On-board Parameter Optimization for Space-based Infrared Air Vehicle Detection based on ADS-B data”. We have revised the manuscript mainly focusing on the journal template and the reference citation error according to your comments.

Attached please find the revised manuscript, in track changes, for your final review and approval.

Sincerely,

Yejin Li, Peng Rao, Zhengda Li, Jianliang Ai

2 June 2023

Reviewer 4 Report

We thank the authors for addressing our previous comments. References in the response however appear to be incorrectly excluded, and might be added to the manuscript text where appropriate.

N/A

Author Response

(The authors gave the same response as above.)
